# A Hierarchical Surface Graph Framework for Protein–Protein Binding Affinity Prediction

**Sharmi Banerjee, Mostafa Karimi**
Amazon
{sharmiba, mkarimii}@amazon.com

**Sukanya Sasmal, Jonah Noh, Pete Bowry**
Amazon
{sukanyaz, jonahnoh, petebow}@amazon.com

**Tommi Jaakkola**
Massachusetts Institute of Technology
tommi@csail.mit.edu

**Bella Dubrov, Shang Shang**
Amazon
{belladub,shashang}@amazon.com

## Abstract

Accurate prediction of protein–protein binding affinity is fundamental to understanding molecular interactions and has broad implications for protein engineering. We introduce **ProtSurf**, a novel protein surface representation learning framework for predicting binding affinity between protein complexes. While many surface-based methods rely only on locally defined neighborhoods, ProtSurf builds a hierarchical surface graph architecture, with a local encoder that aggregates chemical and geometric features from residue-level surface patches and a global graph module that models interactions across the residues. We use graph transformer with relative positional encoding to represent both local and global features and a permutation-invariant reconstruction module to infer node ordering and reconstruct local residue graphs. We further show that data augmentation with protein protein interaction data from AlphaFold improves affinity prediction. Evaluated across three protein–protein affinity tasks — SKEMPI, SAbDab, and HER2 — ProtSurf achieves state-of-the-art performance on these datasets.

## 1 Introduction

Predicting protein–protein binding affinity is a central challenge in computational biology with broad implications for drug discovery, antibody engineering, and protein design. Protein surfaces encode chemical and geometric fingerprints that govern molecular recognition. Surface descriptors have therefore been widely used for downstream tasks such as binding site prediction, protein–ligand interaction modeling, and interface search in protein complexes Gainza et al. (2020). Recent work has increasingly focused on multi-modal protein representation learning to improve binding affinity prediction Cai et al. (2024); Mallet et al. (2023); Wu & Li (2024); Zhang et al. (2024); Somnath et al. (2021). These approaches broadly fall into methods that implicitly encode surface information, such as GearBind Cai et al. (2024), which models all-atom interface graphs, and methods that explicitly represent protein surfaces Mallet et al. (2023); Wu & Li (2024); Somnath et al. (2021). Explicit surface-based models show that surface geometry and physicochemical properties provide complementary information beyond backbone structure alone Zhang et al. (2024).

While local surface features capture the immediate physicochemical environment of an interaction site, their functional interpretation often depends on long-range structural context. Biological phenomena such as allostery—where perturbations at one site influence distant functional regions—demonstrate that global structural dependencies can substantially affect binding and activity beyond local neighborhoods Nussinov et al. (2013). As a result, combining local surface representations with global context is important for accurate affinity prediction. Most surface based methods model protein surface first by extracting local features and then combine other modalities Mallet et al. (2023) to model them as graph neural networks.

We propose **ProtSurf**, a hierarchical surface graph representation framework for binding affinity prediction. ProtSurf combines a local surface graph encoder that processes residue-level physicochemical descriptors computed from MaSIF Gainza et al. (2020) with a global protein graph encoder that

models quantized residue embeddings using backbone topology. The globally contextualized residue embeddings are decoded using a local residue graph decoder together with a permutation inference module Winter et al. (2021). Both encoders are implemented as graph transformers with relative positional encodings Park et al. (2022). This design preserves fine-grained interaction specificity while incorporating long-range structural context.

We evaluate ProtSurf on three protein–protein binding affinity benchmarks: SKEMPI Jankauskaitė et al. (2019), HER2 Shanehsazzadeh et al. (2023), and SAbDab Dunbar et al. (2014), covering both binding energy prediction and mutation-induced affinity change estimation. ProtSurf outperforms other binding affinity prediction models on the first two datasets and is a runner up on SAbDab dataset highlighting the value of the combined hierarchical representation as an inductive bias for protein structure–function modeling.

Our contributions are as follows:

- We introduce a hierarchical surface graph architecture that integrates local surface physicochemical features with global protein backbone context using graph transformers with relative positional encodings.

- We show that increasing model capacity and pre-training data augmentation improves the quality and transferability of learned surface representations.

- We demonstrate competitive or superior performance to state-of-the-art methods across multiple protein–protein binding affinity benchmarks.

## 2 METHODS

We propose ProtSurf, a framework that encodes surface residues using local context-aware features and updates them via protein backbone information. By incorporating permutation-invariant reconstruction, ProtSurf effectively aligns nodes within the residue graph, enabling the decoder to accurately recover both node features and adjacency matrices.

### 2.1 DATA PROCESSING

We describe the featurization of surface residues for the local graph encoder and the protein graph for the global decoder. Training data for each protein consists of a dataset with both local and global features in the form of a dictionary.

#### 2.1.1 RESIDUE GRAPH REPRESENTATION FROM LOCAL NEIGHBORHOOD

Given a protein structure, each residue is represented based on its local neighborhood in structural space, explicitly computing its chemical and geometric properties. We process each protein structure using the MaSIF protocol Gainza et al. (2020). MaSIF decomposes a surface into overlapping radial patches (which consist of three vertices) with a fixed geodesic radius, where each vertex is assigned five features: electrostatic potential (charge), hydrophobicity, hydrogen bond interaction propensity, shape index, and distance-dependent curvature. In addition, we introduce two geometric features: (1) the distance from the patch centroid to the residue's $C_\alpha$ atom, and (2) the angle between the vectors from the patch centroid to $C_\alpha$ and from $C_\alpha$ to $C_\beta$. These features are analogous to an anchor point for each patch to the residue. For residues lacking a side chain, we generated a virtual $C_\beta$ atom following the method described in FoldSeek van Kempen et al. (2022). Finally, we averaged the chemical features across three vertices within a patch, resulting in a 5-dimensional chemical feature vector. Combined with the two geometric features, this yields a 7-dimensional feature vector per patch. After mapping patches to residues, we sample 32 patches per residue using Farthest Point Sampling (FPS). Priority is given to patches closely associated with the residue, followed by patches containing neighboring side-chain atoms, and finally patches containing only hydrogen atoms from its own or neighboring residue. When reconstructing the node features we weighed the nodes according to this classification. Since patches in a residue do not have any order we adopt relative positional encoding Park et al. (2022) approach to represent the patches as a residue graph. To that end we need two additional matrices, a shortest path distance matrix and a edge type matrix. We construct both of these tensors per residue using distance thresholds as described in Appendix A, B

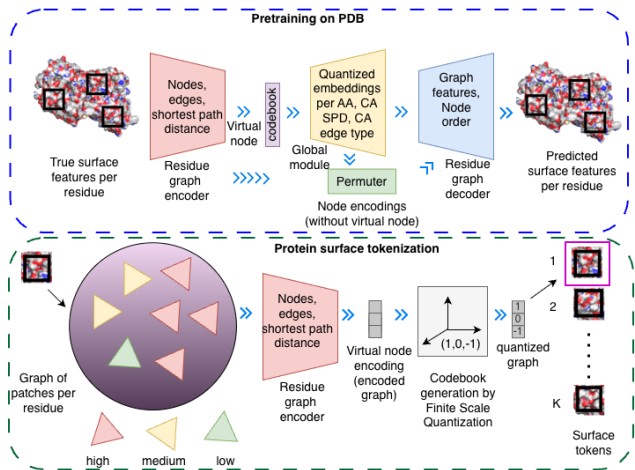

Figure 1: ProtSurf architecture. Local surface residue graphs are encoded, quantized, and updated via a global protein-level module using backbone information.

### 2.1.2 PROTEIN GRAPH REPRESENTATION FROM BACKBONE

Similar to creating graphs per residue, we create graphs for a given protein by using its backbone information. This is required to help learn the global protein geometry via a coarse approximation. We need two matrices for the global context similar to the residue graph step: shortest path distance matrix and edge type matrix which are created from the $C_\alpha$ atom of each surface residue. We create the shortest part distance matrix and the edge type matrix for the global modules using C-$\alpha$ atoms of for each residue. We defined edge types based on 4Å, 8Åand 12Å. For example, for a given protein of 100 surface exposed residues there will be 100 graphs processed by the local encoder in parallel and 1 graph processed by the global module.

### 2.2 MODELING

The main components of the model are (1) a local graph encoder $g_{enc}$ that operates on residue graphs, (2) a finite scale quantized protein surface tokenizer (FSQ), (3) a global decoder which has the same architecture as local encoder but operate on protein graphs, and (4) a local graph decoder that reconstructs the residue graphs and (5) *permuter* $p_{perm}$ to learn the alignment between the input graph and the reconstructed one. We illustrate the overall pipeline in Figure 1 and each of these components in the following subsections.

### 2.2.1 RESIDUE GRAPH ENCODER

We use the graph transformer with learnable relative positional encodings (GRPE) proposed by Park et al. (2022). Node updates are computed via full self-attention augmented with learnable topological and edge-type embeddings derived from shortest-path distances and discretized edge categories. This formulation preserves graph structure without linearization. Given two nodes $n_i$ and $n_j$, the core idea is to represent the residue graph via two additional attention terms as follows: Topological relationship between nodes $n_i$ and $n_j$:

$$a_{(i,j)}^{topology} = q_i \mathcal{P}_{\psi(i,j)}^{query} + k_i \mathcal{P}_{\psi(i,j)}^{key} \tag{1}$$

Edge relationship between nodes $n_i$ and $n_j$:

$$a_{(i,j)}^{edge} = q_i \mathcal{E}_{(i,j)}^{query} + k_i \mathcal{E}_{(i,j)}^{key} \tag{2}$$

For details please see Appendix C.

### 2.2.2 FINITE SCALE QUANTIZER

We adopted Finite Scale Quantization Mentzer et al. (2023) to create a protein surface codebook. For details see Appendix D

### 2.2.3 PROTEIN GRAPH DECODER

To incorporate protein-level structural context prior to reconstructing local residue features, we introduce a global module positioned between the surface quantizer and the local decoder. The global module shares the same architecture as the local graph encoder but operates on a different graph definition: instead of one graph per residue, it processes a single graph per protein, with nodes corresponding to quantized embeddings of surface-exposed residues. This protein-level graph is constructed using backbone geometry, enabling information exchange across residues through the protein structure. Padding and dynamic masking are employed to support proteins of variable length. The resulting globally updated, quantized residue embeddings are expanded to match the local graph sizes (32 patches per residue) and passed to the local decoder for reconstruction. We provide two other alternative global–local integration architectures in Appendix E

### 2.2.4 RESIDUE GRAPH DECODER

Since the graph encoding from the global module is in $d_{\text{global}}$, we used a simple linear layer to project it back to the $d_z$ embedding that serves the purpose of node features for the decoder denoted $z_{\text{dec}}$. Inspired by Winter et al. (2021), we defined sinusoidal positional embedding $\text{PE} \in \mathbb{R}^{N \times d_z}$ with the i-th node's embedding for k-th dimension as follows:

$$\text{PE}(i)_k = \begin{cases} \sin(i/10000^{2k/d_z}), & \text{if k is even} \\ \cos(i/10000^{2k/d_z}), & \text{k is odd} \end{cases} \tag{3}$$

Then we used the learned permutation matrix $\hat{\mathbf{P}}$ to reorder the positional embedding by multiplication $\text{PE}_{\text{update}} = \hat{\mathbf{P}} \times \text{PE}$. Finally, we concatenated the node features of the decoder with the updated positional encoding and passed them to the graph decoder. The graph decoder exactly follows the graph encoder with minor differences at the final project layers:

$$z_o = g_{\text{dec}}([z_{\text{dec}} \,||\, (PE)_{\text{update}}])$$
$$\hat{m}_{\text{node}} = W_{\text{node}} z_o + b_{\text{node}} \tag{4}$$
$$\hat{m}_{\text{edge}} = W_{\text{edge}} z_o + b_{\text{edge}}$$

where $\hat{m}_{\text{node}}$ is used to reconstruct the initial node features $m_{\text{node}}$ and $\hat{m}_{\text{edge}}$ is used to reconstruct the un-directed adjacency matrix $\mathbf{A}_\pi$.

### 2.2.5 PERMUTER

Residue-level surface graphs do not admit a canonical node ordering, making direct reconstruction sensitive to permutations. To address this, we incorporate a permutation-invariant alignment module inspired by Winter et al. (2021). Given decoded node embeddings and ground-truth residue features, the permuter learns a soft correspondence matrix that aligns predicted nodes to reference residues, enabling reconstruction losses to be computed without assuming a fixed ordering. The alignment is optimized jointly with the reconstruction objective. The permuter learns to transform/permute this canonical order to a given input node order. For each node $i$ of the input graph, the permuter predicts a score $s_i$ corresponding to its probability of having a low node index in the decoded graph. By sorting the input nodes indices by their assigned scores, we inferred the output node order and constructed the corresponding permutation matrix $\mathbf{P}_{\pi \to \pi'} = (p_{ij}) \in \{0, 1\}^{n \times n}$ with

$$p_{ij} = \begin{cases} 1, & \text{if } j = \text{argsort}(s)_i \\ 0, & \text{else} \end{cases} \tag{5}$$

to align input and output node order. The argsort operation being non-differentiable, the continuous relaxation of the argsort operator proposed in Prillo & Eisenschlos (2020); Grover et al. (2019) has been used as follows

$$\mathbf{P} \approx \hat{\mathbf{P}} = \text{softmax}(\frac{-d(\text{sort}(s)\mathbf{1}^\top, \mathbf{1}s^\top)}{\tau}) \tag{6}$$

where the softmax operator is applied row-wise, $d(x, y)$ is the $L_1$-norm and $\tau \in \mathbb{R}_+$ a temperature parameter.

Full formulation follows Winter et al. (2021) and is given in Appendix F.

### 2.2.6 LOSSES

We follow the reconstruction objective of Yang et al. (2024), combining node feature reconstruction, adjacency reconstruction, and an entropy regularization term encouraging discrete permutations.

Following (Yang et al., 2024) we define node and edge reconstruction as

$$\mathcal{L}_{\text{rec}} = \frac{1}{w_i} \sum_{i=1}^{N} (1 - \frac{m_{\text{node}}^T \hat{m}_{\text{node}}}{||m_{\text{node}}||.||\hat{m}_{\text{node}}||} + ||\mathbf{A}_\pi - \sigma(\hat{m}_{\text{edge}}.\hat{m}_{\text{edge}}^T)||^2) \tag{7}$$

where $\sigma(.)$ is the sigmoid function and $w_i$ is the weight of each node (residue) normalized by the length of the protein. In order to push the *soft* permutation matrix towards a real permutation matrix (i.e. contains one 1 in every row and column), we introduce an additional penalty term to minimize the Shannon entropy both row-wise and column-wise:

$$\text{C}(\hat{\mathbf{P}}) = \sum_i \text{H}(\bar{\mathbf{p}}_i) + \sum_j \text{H}(\bar{\mathbf{p}}_j) \tag{8}$$

with Shannon entropy $\text{H}(x) = -\sum_i x_i \log(x_i)$ and normalized probabilities $\bar{\mathbf{p}}_i, = \frac{\hat{\mathbf{P}}_i}{\sum_j \hat{\mathbf{P}}_{i,j}}$. The final loss would be:

$$\mathcal{L} = \mathcal{L}_{\text{rec}} + \lambda \text{C}(\hat{\mathbf{P}}) \tag{9}$$

where $\lambda$ hyper-parameter would balance between main reconstruction loss and the additional penalty.

## 3 EXPERIMENTS

We first pre-train ProtSurf on diverse protein interfaces to reconstruct local residue graphs. We then extract per-residue embeddings from the pretrained encoder and use them as inputs to downstream affinity prediction models. Following Wu & Li (2024), we evaluate ProtSurf on three protein–protein binding affinity prediction tasks using the SKEMPI Jankauskaitė et al. (2019), HER2 Shanehsazzadeh et al. (2023), and SAbDab Dunbar et al. (2014) datasets.

### 3.1 PRE-TRAINING DATA CONSTRUCTION

We construct a diverse pre-training corpus by augmenting experimentally validated protein structures from the RCSB database Burley et al. (2023) with predicted interface configurations from AlphaFold predictions Varadi et al. (2024). Specifically, we included (i) full protein complexes from RCSB (approximately 200k structures), (ii) single-chain decompositions of multi-chain complexes, yielding approximately 440k individual chains after filtering validation overlaps, and (iii) interacting protein domains extracted from AlphaFold-predicted structures using domain annotations. This combination exposes the model to a wide range of interface geometries and interaction contexts. For details see Appendix G.

### 3.2 PRE-TRAINING PROTSURF

We train two model variants with 10M and 60M parameters using learning rates of $2 \times 10^{-4}$ and $7 \times 10^{-5}$, respectively. Training used the AdamW optimizer with weight decay 0.1, a per-GPU batch size of 2, and a global batch size of 128 across 8 NVIDIA H200 GPUs for 100 epochs. We adopt the implementation of the graph encoder module from the GRPE GitHub Park et al. (2022), the FSQ

Table 1: Evaluation of mutant effect prediction ($\Delta\Delta G$) on the SKEMPI.v2 dataset. Best results are highlighted in **bold**, and second best results are underlined.

| CATEGORY | METHOD | PEARSON | SPEARMAN | RMSE | MAE |
|---|---|---|---|---|---|
| ENERGY FUNCTION | ROSETTA | 0.3113 | 0.3468 | 1.6173 | 1.1311 |
| | FOLDX | 0.3120 | 0.4071 | 1.9080 | 1.3089 |
| SEQUENCE-BASED | ESM-1V | 0.1921 | 0.1572 | 1.9609 | 1.3683 |
| | PSSM | 0.0159 | 0.0666 | 1.9978 | 1.3895 |
| | MSA TRANSF. | 0.1173 | 0.1313 | 1.9835 | 1.3816 |
| | TRANCEPTION | 0.1141 | 0.1402 | 2.0382 | 1.3883 |
| SUPERVISED | DDGPRED | 0.6580 | 0.4687 | 1.4998 | 1.0821 |
| | END-TO-END | 0.6373 | 0.4882 | 1.6198 | 1.1761 |
| | GEARBIND | 0.6590 | 0.4980 | 1.6390 | 1.1430 |
| PRE-TRAINING-BASED | ESM-IF | 0.3194 | 0.2806 | 1.8860 | 1.2857 |
| | B-FACTOR | 0.2390 | 0.2625 | 2.0411 | 1.4402 |
| | MIF-NET. | 0.6523 | 0.5134 | 1.5932 | 1.1469 |
| | RDE-NET. | 0.6447 | 0.5584 | 1.5799 | 1.1123 |
| | PPIFORMER | 0.6450 | 0.5304 | 1.6420 | 1.1186 |
| | SURFACE-VQMAE | 0.6482 | 0.5611 | 1.5876 | 1.1271 |
| | PROMPT-DDG | 0.6772 | **0.5910** | 1.5207 | **1.0770** |
| | ATOMSURF | 0.3592 | 0.3156 | 1.8210 | 1.5090 |
| | GEARBIND+PRETRAINED | 0.6760 | 0.5250 | 1.6110 | 1.1150 |
| OURS | PROTSURF (10M) | 0.6856 | 0.5450 | 1.5525 | 1.1853 |
| | PROTSURF (60M) | **0.7033** | 0.5686 | **1.4969** | 1.1402 |

quantizer part from Lucidrains GitHub Wang (2024) and the permutation invariant part from Winter et al. (2021) GitHub. We follow the FSQ paper Mentzer et al. (2023) to select different vocabulary sizes and hidden dimensions. We also implement adaptive RMSNorm from Ma et al. (2025) and scale the attention weights to stabilize training.

## 3.3 AFFINITY PREDICTION DATA

We evaluate ProtSurf on three benchmark datasets: (1) SKEMPI v2 Jankauskaitė et al. (2019) for mutation-induced binding free energy change prediction ($\Delta\Delta G$), (2) HER2 Shanehsazzadeh et al. (2023) for protein–protein binding affinity prediction, and (3) SAbDab Dunbar et al. (2014) for antibody–antigen binding free energy prediction ($\Delta G$).

For predicting change in binding affinity $\Delta\Delta G$ prediction on SKEMPI v2, we follow Luo et al. (2023) and report Pearson and Spearman correlation coefficients, root mean squared error (RMSE), mean absolute error (MAE). SKEMPI Jankauskaitė et al. (2019) contains data on changes in the thermodynamic parameters and kinetic rate constants after mutation for structurally resolved PPIs. The latest version contains manually curated binding data for 7,085 mutations. The dataset is split into three folds by structure, each containing unique protein complexes that do not appear in other folds. Two folds are used for train and validation, and the remaining one is used for test. This yields three different sets of parameters and ensures that every data point in SKEMPI.v2 is tested once.

For HER2 Shanehsazzadeh et al. (2023) we report Pearson correlation on $\Delta\Delta G$. This dataset contains high-quality binding affinity data, measured by surface plasmon resonance (SPR) on 419 HER2 binders with de-novo designed CDR loops. The antibodies in the dataset are variants of Trastuzumab that have high edit distance (7.6 on average), making them potentially challenging for $\Delta\Delta G$ predictors trained on low-edit-distance data.

For absolute binding affinity prediction tasks ($\Delta G$) on SAbDab, we report Pearson and Spearman correlation between predicted and experimentally measured binding free energies. SAbDab Dunbar et al. (2014) contains 4,883 antibody-antigen complexes after removing duplicates and structures without antigens. Among them, 566 instances have binding affinity labels and are used as the test set. Following Myung et al. (2022) we create a training set with 197 complexes after removing instances appeared in the test set. Then we randomly split them into two halves with a validation ratio of 50%.

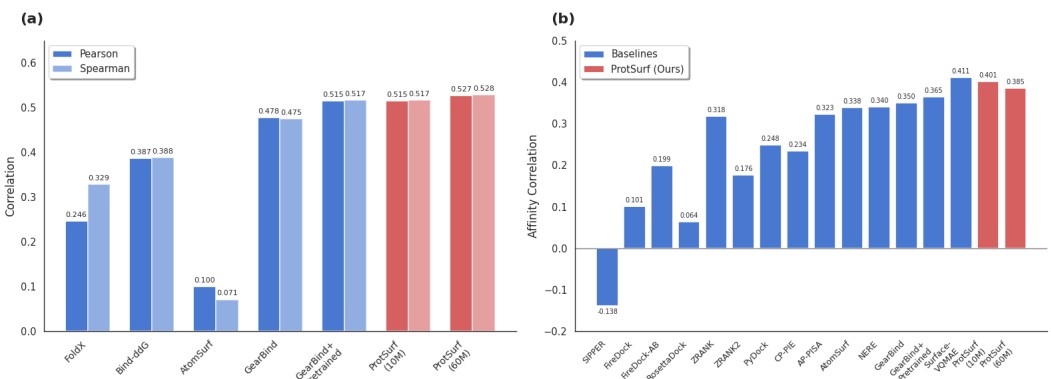

Figure 2: Affinity prediction

## 3.4 DOWNSTREAM EVALUATION OF PROTSURF ON AFFINITY PREDICTION

We leverage GearBind Cai et al. (2024) to evaluate ProtSurf on predicting $\Delta\Delta G$ for SKEMPI Jankauskaitė et al. (2019) and HER2 Shanehsazzadeh et al. (2023), and $\Delta G$ on SAbDab. We extract per-residue embeddings from pre-trained ProtSurf and augment them with one-hot residue encodings in the protein graph during dataset preparation for GearBind. We use the pre-trained GearBind model and fine-tune it with augmented features using the same architecture and configurations defined in their paper. This design choice is motivated by prior evidence that surface representations provide complementary information beyond backbone structure Mallet et al. (2023); Zhang et al. (2024).

## 4 RESULTS

### 4.1 PERFORMANCE ON MUTATION-INDUCED BINDING AFFINITY CHANGE PREDICTION (SKEMPI)

In Table 1 we compare the performance of ProtSurf on mutation-induced binding affinity change prediction ($\Delta\Delta G$) on the SKEMPI v2 dataset. ProtSurf (60M) achieves state-of-the-art performance on Pearson correlation metrics (0.7033) and root mean square error (1.4969). The 10M-parameter variant remains highly competitive ranking second in Pearson correlation (0.6856), indicating that the proposed surface-based representation learning framework is effective even at modest model capacity Appendix H. Compared to other surface-centric methods such as Surface-VQMAE and AtomSurf, ProtSurf consistently improves correlation metrics, suggesting that jointly modeling local surface geometry with global structural context provides more informative representations for predicting mutation-induced binding affinity changes.

### 4.2 EVALUATION ON THE HER2 BINDERS TEST SET

As shown in Figure 2a, ProtSurf achieves the strongest Pearson and Spearman correlation across methods on zero shot prediction of $\Delta\Delta G$ on HER2 dataset, outperforming both classical energy-based approaches and local surface models. This result highlights the ability of hierarchical surface surface representations to generalize to structurally diverse antibody variants.

### 4.3 ANTIBODY–ANTIGEN BINDING AFFINITY PREDICTION ON SABDAB

Figure 2b compares ProtSurf against classical docking and modern surface-based methods on SAbDab. ProtSurf achieves strong correlation and improves over several established baselines, while remaining competitive with SurfaceVQMAE Wu & Li (2024). We noticed that for SAbDab the smaller 10M ProtSurf version outperforms the larger 60M version. This could be due to the small size of the dataset itself and we hypothesize this task could benefit from data augmentation.

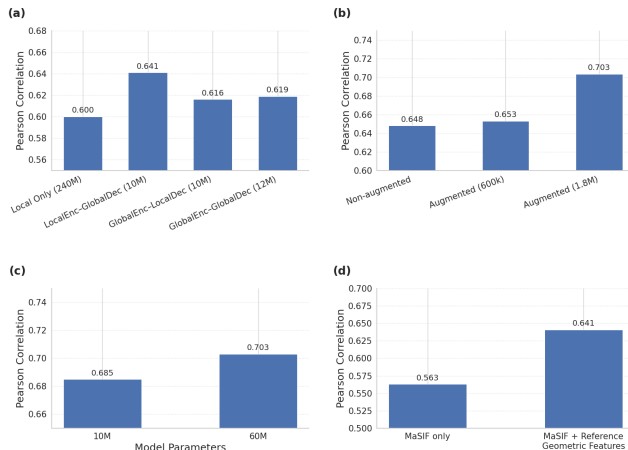

Figure 3: Ablation studies: (a) architecture variants with and without global context, (b) effect of data augmentation on pre-training loss for the 60M ProtSurf model, (c) SKEMPI performance of 10M vs. 60M ProtSurf pre-trained on 200k proteins, (d) impact of reference geometric features on SKEMPI.

## 5 ABLATION STUDIES

We first compare architectural variants that introduce the global module at different stages, alongside a purely local baseline (Figure 3a). All models are pre-trained on 200k RCSB protein complexes and evaluated on SKEMPI without the pre-trained Gearbind model. Adding global module only at decoding (Local Encoder–Global Decoder) substantially improves performance over the local baseline (Pearson $0.600 \rightarrow 0.641$). In contrast, applying global context earlier—such as Global Encoder–Local Decoder—results in weaker performance (Pearson 0.616), and fully global variants do not yield additional gains (Pearson 0.619). These results suggest that introducing global information prior to quantization may attenuate its benefits, as long-range dependencies can be lost during quantization. We next analyze the impact of scaling pre-training data and data augmentation. Increasing both model capacity and training data consistently reduces validation loss, with models trained on augmented datasets achieving lower reconstruction error than those trained without augmentation at the same scale. This indicates that generating diverse interface views by decomposing protein complexes provides a stronger inductive signal for learning transferable surface representations. A similar trend is observed in downstream $\Delta\Delta G$ prediction performance on SKEMPI (Figure 3b). Next, we examine the effect of model size by comparing 10M and 60M parameter variants. Scaling from 10M to 60M parameters leads to consistent improvements in both pre-training and downstream tasks: validation loss decreases from $0.0545 \rightarrow 0.053$ on 200k proteins and from $0.0517 \rightarrow 0.0509$ on 600k proteins, while downstream Pearson correlation improves from $0.685 \rightarrow 0.703$ on AlphaFold proteins (Figure 3c). Finally, we assess the contribution of the two additional geometric features introduced on top of the standard MaSIF inputs. Pre-training a Local Encoder–Global Decoder model using only the original five MaSIF features results in a substantial degradation in performance, with Pearson correlation dropping from 0.649 to 0.563 (Figure 3d), highlighting the importance of these features for accurate interface representation.

## CONCLUSION

Recent structure-based approaches for predicting protein–protein binding affinity have achieved strong performance but largely focus on atomic or micro-environmental features, leaving explicit surface-level representations underexplored. We presented ProtSurf, a hierarchical framework that captures local geometric features and global inter-residue relationships through protein surface embeddings. Our results show that integrating surface-aware representations with existing models like GearBind enhances binding affinity prediction. This confirms that surface-level features provide critical, complementary information to atomic models, offering a promising path for future research in mutation effect prediction and antibody design.

MEANINGFULNESS STATEMENT

Proteins are fundamental to life, acting as essential building blocks for structure, growth, and repair in every cell. As crucial functional molecules, they act as enzymes to accelerate chemical reactions, hormones to send signals, and antibodies to defend the immune system. In this work we integrate protein sequence, structure, and surface modalities to build representations that are predictive for downstream binding affinity prediction.

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

## A  LOCAL SURFACE FEATURIZATION DETAILS

Protein surfaces are computed from atomic structures using MSMS to generate a triangulated solvent-excluded surface. Surface patches are extracted following the MaSIF framework (Gainza et al., 2020) by selecting surface vertices within a geodesic radius of 9 Å around each patch center. Each patch is parameterized in local polar coordinates and discretized on a fixed grid.

For each surface patch, we compute a set of physicochemical features including shape index, mean curvature, electrostatic potential (computed using APBS), hydrogen bond donor and acceptor indicators, and hydrophobicity. These features are concatenated to form a patch-level descriptor vector.

We modified the MaSIF processing code to output a vertex-to-residue mapping, enabling accurate feature computation for each surface-exposed residue. For a given residue A, we first selected all patches that are within 3Å from any of its atoms. As not all of the patches will be mapped to the residue (as per MaSIF calculation), we then categorized the patches into three groups: (1) *Core* — patches where all three vertices map to atoms from residue A, (2) *Border* — patches where at least one vertex maps to an atom from another residue, and (3) *Borrowed* — patches where no vertices map to atoms from residue A. Borrowed patches are filtered out for residue A. We assigned a label to each patch: (1) *High* — if its closest atoms include $\{C, O, N, S\}$ or all heavy atoms from its

assigned residue, (2) *Medium* — if its closest atoms include $\{C, O, N, S\}$ or all heavy atoms from neighboring residue , and (3) *Low* — for patches whose closest atoms are all hydrogen atoms. We represented each residue as a graph $G = (V, E)$, where the nodes $V = \{n_i\}_{i=1:N}$ correspond to $N$ randomly sampled patches following 70% from high, 20% from medium and 10% from low patches. We chose $N$ as 32 following MaSIF Gainza et al. (2020) in their tasks and chose majority of the patches that are most closely tied to the residue, followed by those that contain side-chain atoms of the neighboring residues and finally the last fraction consisting of only hydrogen atoms from its own or neighboring residue. When reconstructing the node features we weighed the nodes according to this classification. The above-mentioned 7-dimensional features are used as node features. The patch types are illustrated in Figure 4.

The edges $E = \{e_{ij} | j \in \mathcal{N}_i\}_{i=1:N}$ are defined where $\mathcal{N}_i = \{j | \mathrm{dist}(n_i, n_j) < 3\text{Å}\}$ is a set of neighbors of a node $n_i$ and $\mathrm{dist}(., .)$ is defined as the distance between the patch centroids of nodes $n_i$ and $n_j$. In addition, a virtual node has been added to the graph that is connected to every other node in the graph through its special virtual edge. This virtual node will serve a similar purpose as the [CLS] token in transformers. Since the [CLS] token has been commonly utilized to provide sentence embedding, we used the virtual node to calculate the final graph embedding and its tokenized representation. Edges are featurized according to the centroid distances: (1) *Short* — where their distance is less than 1Å , (2) *Medium* — where their distance is between 1Å and 2Å , (3) *Long* — where their distance is between 2Å and 3Å (4) *Virtual* — edge between virtual node and any other node, (5) *Self* — edge connecting each node to itself, and (6) *No* — nodes that are not connected to each other. Therefore, there are six categories of edges. Inspired by (Park et al., 2022), we have used the topological relationship $\psi(i, j)$ between nodes $n_i$ and $n_j$ based on their shortest path distance with the maximum cutoff *max-hop*. Formally $\psi(i, j)$ is featurized as following: (1) *Unreachable* — No connection between two nodes, (2) *shortest path distance s* — Shortest path distance value $s \in \{0, \cdots, \text{max-hop}\}$ between nodes $n_i$ and $n_j$, (3) *Far distance* — If the shortest path distance is greater than *max-hop*, and (4) *Virtual* — edge between virtual node and any other node. Therefore, there are *max-hop* + 4 categories of topological relation.

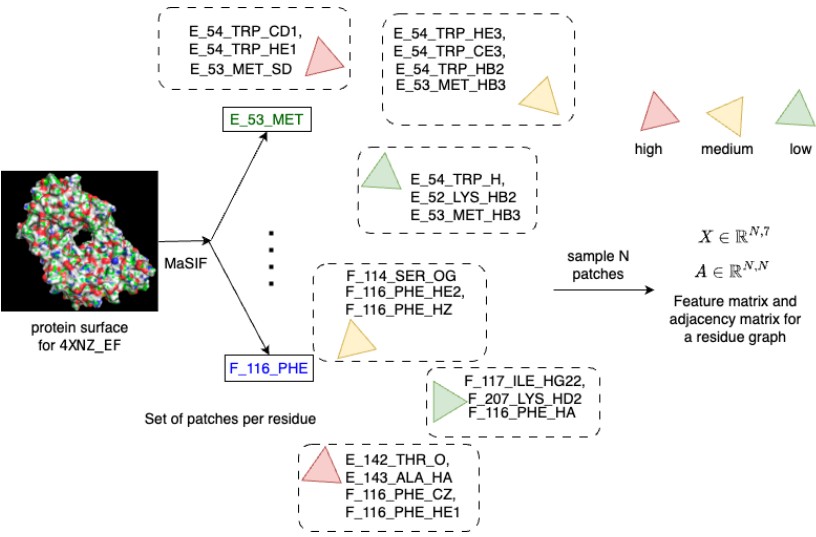

Figure 4: Creating local geometric and chemical features per residue: Given a protein structure, we first run MaSIF and get 5 features mapped to each surface exposed residue: charge, hydrophobicity, shape index, distance dependent curvature, hydrogen bond interaction. We compute 2 more features: patch centroid to C-$\alpha$ atom of residue and angle between C-$\alpha$ to patch centroid and C-$\alpha$ to C-$\beta$. The patches are classified as high medium or low depending on the type of core or border or borrowed atoms

# B    LOCAL AND GLOBAL GRAPH METRICS

We selected the edge thresholds for both local Figure 6 and global residue graphs Figure 5 by sampling 500 random single and multi chain proteins from the RCSB database and computing metrics such as community sizes, no. of different edges and selected parameters that led to decent community sizes and edge types.

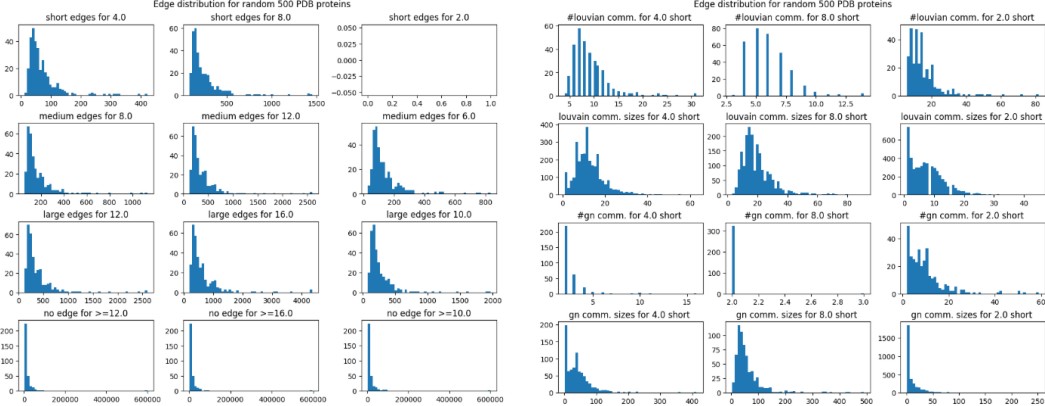

Figure 5: Graph metrics for global protein graphs

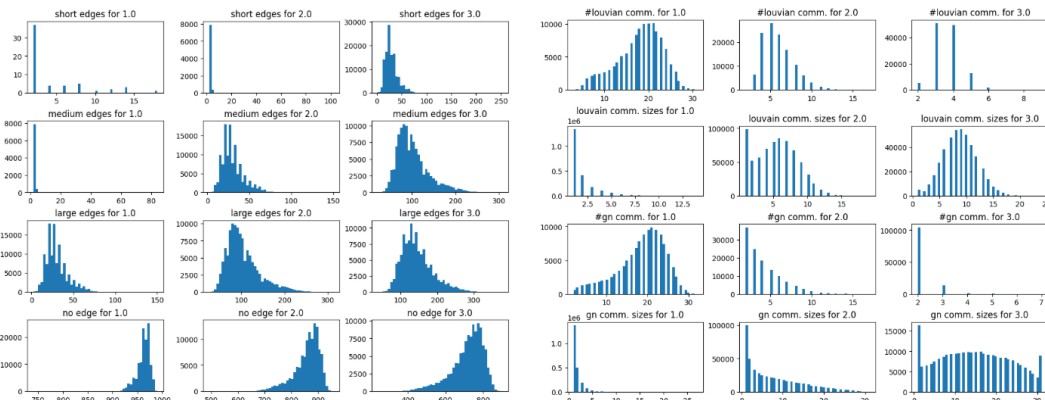

Figure 6: Graph metrics for local residue graphs

## C    RESIDUAL GRAPH ENCODER

We use the graph transformer with learnable relative positional encoding developed by (Park et al., 2022), which uses (1) dot-product attention commonly used in transformers, (2) learnable topological relationship $\mathcal{P}_{\psi(i,j)} \in \mathbb{R}^{d_z}$ between nodes $n_i$ and $n_j$, and (3) learnable edge relationship $\mathcal{E}_{(i,j)} \in \mathbb{R}^{d_z}$ between nodes $n_i$ and $n_j$. Let us assume $x_i \in \mathbb{R}^{d_x}$ denotes the input feature of the node $n_i$ with $d_x$ as its dimension, and $z_i \in \mathbb{R}^{d_z}$ denotes the final output feature of transformer's layer with $d_z$. First, the self-attention module computes query $q_i$, key $k_i$, and value $v_i$ with independent linear transformations $W^{\text{query}} \in \mathbb{R}^{d_x \times d_z}$, $W^{\text{key}} \in \mathbb{R}^{d_x \times d_z}$ and $W^{\text{value}} \in \mathbb{R}^{d_x \times d_z}$.

$$q_i = W^{\text{query}} x_i, k_i = W^{\text{key}} x_i \; v_i = W^{\text{value}} x_i \tag{10}$$

Second, the topological relationship between nodes $n_i$ and $n_j$ is calculated as:

$$a_{(i,j)}^{\text{topology}} = q_i \mathcal{P}_{\psi(i,j)}^{\text{query}} + k_i \mathcal{P}_{\psi(i,j)}^{\text{key}} \tag{11}$$

Next, the edge relationship between nodes $n_i$ and $n_j$ is calculated as:

$$a_{(i,j)}^{\text{edge}} = q_i \mathcal{E}_{(i,j)}^{\text{query}} + k_i \mathcal{E}_{(i,j)}^{\text{key}} \tag{12}$$

Finally, the overall attention map is computed summing these three terms. Attention here denotes full pairwise attention between the nodes adjusted by the graph features from the two additional matrices.

$$
\begin{aligned}
a_{(i,j)} &= \frac{q_i.k_j + a_{(i,j)}^{\text{topology}} + a_{(i,j)}^{\text{edge}}}{\sqrt{d_z}}, \\
\hat{a}_{(i,j)} &= \frac{\exp(a_{(i,j)})}{\sum_{k=1}^{N} \exp(a_{(i,k)})}
\end{aligned}
\tag{13}
$$

The overall attention module outputs the next hidden feature by applying weighted summation on the values

$$z_i = \sum_{j=1}^{N} \hat{a}_{(i,j)} v_j \tag{14}$$

The utilized learnable relative positional encoding can be seen as an alterative to linearizing graphs, thus enabling richer node-topology and node-edge interactions since it preserves structural graph information.

## D    FINITE SCALE QUANTIZER

We adopted Finite Scale Quantization Mentzer et al. (2023) to create a protein surface codebook. FSQ creates a simple, fixed grid partition in a lower-dimensional space. Let us assume the FSQ's internal dimension is represented as $d_{\text{FSQ}}$ and the $i^{th}$ dimension can have $L_i$ different integers or *levels*. Therefore, overall *implicit* codebook size for FSQ with $\{L_1, \cdots, L_{d_{\text{FSQ}}}\}$ can be $|\mathcal{C}| = \prod_{i=1}^{d_{\text{FSQ}}} L_i$. FSQ module takes in the virtual node of a residue graph from the encoder $z_{\text{graph}} \in \mathbb{R}^{d_z}$, down-projects the graph representation down to $d_{\text{FSQ}}$ dimension through $z_{\text{latent}} = \text{MLP}(z_{\text{graph}}) \in \mathbb{R}^{d_{\text{FSQ}}}$. Then, non-differentiable online quantization occurs for each dimension $i$ through $z_{\text{FSQ},i} = \text{round}(\lfloor L_i/2 \rfloor \tanh(z_{\text{latent},i}))$. The quantization step binds the encoder output to $L$ values, which is the number of dimensions of the quantizer, and then rounds to integers, leading to a quantized codebook. Since $\text{round}(.)$ function is a non-differential operation, straight-through estimator (STE) Bengio et al. (2013) can be used to propagate gradient through $\text{round\_ste}(x) = x + \text{stop\_gradient}(\text{round}(x) - x)$.

## E  PROTSURF ARCHITECTURE VARIATIONS

We propose three variations of the ProtSurf based on position of the global module and a local ProtSurf version in the sections below. The main differences among the three is the position of the global module that leads to getting surface tokens that are either global aware or local but the feature reconstruction in all cases is global aware.

**Local Surface Embedding with Global-Aware Reconstruction**    Virtual node embeddings are first quantized using Finite Scale Quantization (FSQ) to produce quantized tokens and their corresponding quantized embeddings. These quantized embeddings are then processed by a global module operating on a single protein-level graph defined by $C_\alpha$–$C_\alpha$ SPD and edge types. Padding and dynamic masking are applied to support variable protein lengths. The globally updated quantized embeddings are merged with local node embeddings via a permuter and passed to the local decoder Figure 7

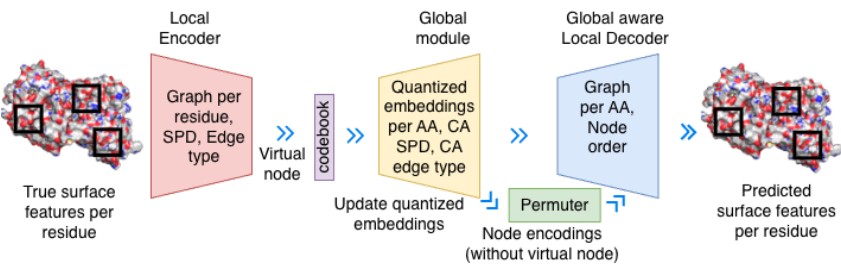

Figure 7: Local Encoder Global Decoder

**Global Surface Embedding with Local Reconstruction**    Virtual node embeddings are first updated by the global module at the protein level and subsequently quantized using FSQ. The resulting globally aware quantized embeddings are combined with local node embeddings and passed to the decoder, yielding globally informed but locally reconstructed surface features Figure 8.

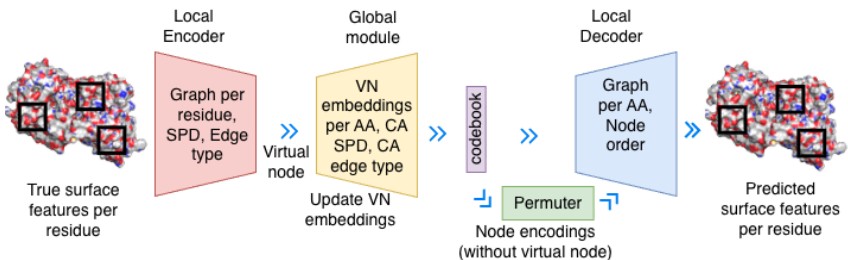

Figure 8: Global Encoder Local Decoder

**Global Surface Embedding with Global-Aware Reconstruction**    This hybrid variant applies global contextualization both before and after quantization. Virtual node embeddings are globally updated, quantized, and then refined by a second global module. The resulting embeddings are permuted and decoded to produce surface reconstructions Figure 9.

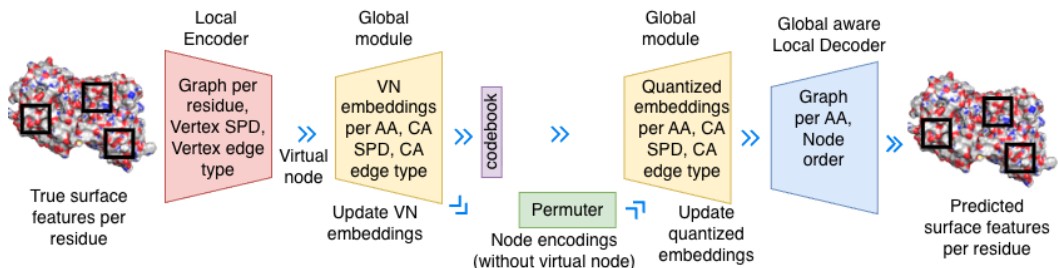

Figure 9: Global Encoder Global Decoder

**Local Surface Embedding** This variant gets rid of the global module altogether. Virtual node embeddings are quantized, and the resulting embeddings are permuted and decoded to produce surface reconstructions Figure 10.

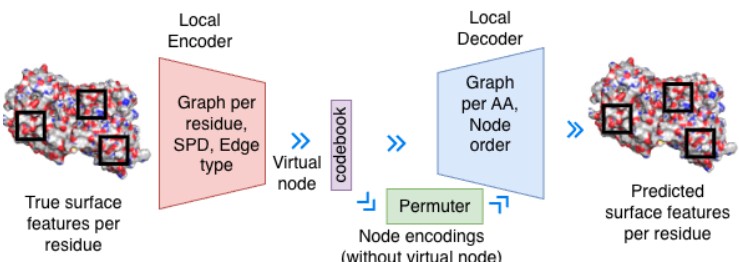

Figure 10: Local Encoder Local Decoder

## F PERMUTATION-INVARIANT RECONSTRUCTION

The permuter enables alignment between the arbitrary input node ordering and the canonical ordering learned by the decoder. The nodes in the original residue graph do not have any positional encoding and their order is arbitrary but fixed during training. Inferring this order during training allows the decoder to align the nodes which would help in the feature loss calculation. Inspired by Winter et al. (2021), we added a permuter module to reconstruct the residue graph with node features and adjacency matrix in ProtSurf. The permuter module learns to align input and output graph through *soft* alignment. Note that in ProtSurf, patches from residues form un-directed graphs with the adjacency matrix $\mathbf{A}_\pi \in \{0,1\}^{n \times n}$ for $n \in N$ in the node order $\pi \in \Pi$ with $\Pi$ is the set of all permutations over $V$. We defined a permutation matrix $\mathbf{P} \in \mathbb{R}^{N \times N}$ that reorders nodes from order $\pi$ to order $\pi'$ as $\mathbf{P}_{\pi \to \pi'} = (p_{ij}) \in \{0,1\}^{n \times n}$ with $p_{ij} = 1$ if $\pi_i = \pi'_j$ and $p_{ij} = 0$ otherwise.

The inputs to the permuter are the node encodings $N \times \mathbb{R}^{d_z}$ obtained from the output of the encoder module. We discard the virtual node at this step and do not try to reconstruct it. The permuter module has to learn how the ordering of nodes in the graph generated by the decoder model will differ from a specific node order present in the input graph. During the learning process, the decoder will learn its own canonical ordering so that, given a latent code $z_{\text{latent}}$, it will always reconstruct a graph in that order. The permuter learns to transform/permute this canonical order to a given input node order. For each node $i$ of the input graph, the permuter predicts a score $s_i$ corresponding to its probability of having a low node index in the decoded graph. By sorting the input nodes indices by their assigned scores, we inferred the output node order and constructed the corresponding permutation matrix $\mathbf{P}_{\pi \to \pi'} = (p_{ij}) \in \{0,1\}^{n \times n}$ with

$$p_{ij} = \begin{cases} 1, & \text{if } j = \text{argsort}(s)_i \\ 0, & \text{else} \end{cases} \tag{15}$$

to align input and output node order. The argsort operation being non-differentiable, the continuous relaxation of the argsort operator proposed in Prillo & Eisenschlos (2020); Grover et al. (2019) has

been used as follows

$$\mathbf{P} \approx \hat{\mathbf{P}} = \mathrm{softmax}(\frac{-d(\mathrm{sort}(s)\mathbf{1}^\top, \mathbf{1}s^\top)}{\tau}) \tag{16}$$

where the softmax operator is applied row-wise, $d(x, y)$ is the $L_1$-norm and $\tau \in \mathbb{R}_+$ a temperature parameter.

## G  PRE-TRAINING DATA

We trained ProtSurf on both experimentally validated protein complexes from the RCSB Burley et al. (2023) and AlphaFold protein structure databases Varadi et al. (2024) as part of data augmentation. In the following sections we describe the data preparation steps.

### G.1  FULL COMPLEXES FROM RCSB

Pretraining is performed using a surface reconstruction objective, where the model learns to encode residue-level surface graphs into discrete tokens and reconstruct local node and edge features from their quantized representations. To encourage generalization across diverse protein interfaces, it is important to train on proteins exhibiting diverse interfacial geometries. As of May 2024, the RCSB database Burley et al. (2023) contains approximately 200K experimentally validated protein structures. We randomly split these into 90% training and 10% validation sets and use them as the basis for pretraining data augmentation.

### G.2  SINGLE CHAINS FROM RCSB COMPLEXES

We start from approximately 200K experimentally validated protein structures provided in the RCSB dataset curated by Mentzer et al. (2023). Among these, approximately 130K structures contain at least two chains. We first extract individual chains from all structures, resulting in approximately 440K augmented single-chain PDBs. We remove any protein at this step that is present in the validation split defined in the previous section to prevent leakage.

### G.3  INTERACTING DOMAIN DATA FROM ALPHAFOLD DATABASE

Protein domains are distinct, self-contained structural and functional units within a protein molecule, typically consisting of 30-200 amino acids that fold independently into stable three-dimensional structures. These modular building blocks often retain their characteristic structure and activity even when separated from the rest of the protein. Interacting protein domains found within a single protein chain can serve as valuable training data augmentation strategy for protein-protein interaction surfaces. We used domain assignments of the AlphaFold Protein Structure Database Varadi et al. (2022) from the Encyclopedia of Domains (TED) database Lau et al. (2024) and extracted approximately 415 million protein-protein interacting surfaces. We randomly sampled 1 million samples for pre-training.

## H  PEARSON VISUALIZATION ON SKEMPI

Scatter plots of true vs. predicted $\Delta\Delta G$ for 10M and 60M ProtSurf versions.

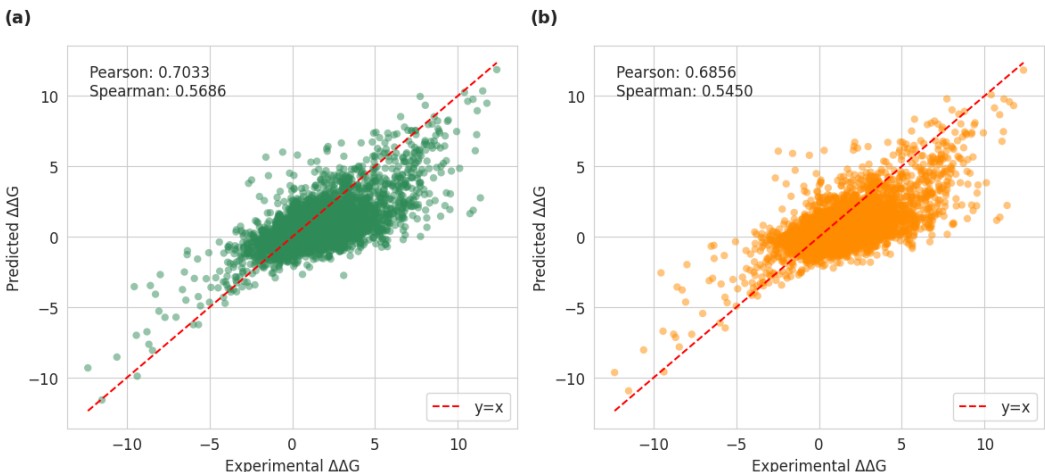

Figure 11: Pearson-R of $\Delta\Delta G$ by 60M ProtSurf (left) and 10M ProtSurf (right).

