# OpenReview forum: "A Hierarchical Surface Graph Framework for Protein–Protein Binding Affinity Prediction"
_ICLR.cc/2026/Workshop/LMRL — ICLR 2026 Workshop LMRL Poster_

### Official Review · Reviewer_4APB · 2026-02-12
**Hierarchical Surface Graph Learning for Binding Affinity Prediction**

**Rating:** 7
**Confidence:** 3

**Review:**

This paper proposes ProtSurf, a hierarchical surface-based graph autoencoder for learning residue-level embeddings. The model encodes local surface patch graphs, incorporates global protein context via a residue-level graph module, and uses the learned embeddings in combination with a fine-tuned GearBind model for binding affinity prediction. The method is evaluated on binding energy prediction and mutation-induced affinity change estimation across three datasets:SKEMPI, HER2, and SAbDab.

The methodology is clearly described and technically detailed. The hierarchical design—combining local surface geometry with global backbone context—is well motivated. The ablation studies are comprehensive and help justify architectural choices. Empirically, the model achieves strong performance on multiple benchmarks.

Weaknesses / Questions:
- The motivation for the autoencoder-based pretraining objective could be better justified. It is unclear whether reconstruction of surface graphs is the most effective pretraining signal for affinity prediction. A comparison with a fully end-to-end supervised variant (training the same architecture directly on binding affinity) would clarify the necessity and benefit of the autoencoder pretraining.
-  The finite-scale quantizer introduces a discrete bottleneck. It would be helpful to analyze how this affects the learned representations and whether it leads to information loss that could impact downstream performance. An ablation comparing with a continuous latent space would be informative.

Overall, this is a technically solid and well-executed paper with strong experimental results, but further justification of the pretraining strategy would strengthen even more the contribution.

---

### Official Review · Reviewer_4WnU · 2026-02-24
**Solid engineering work on hierarchical surface representation for protein-protein affinity prediction**

**Rating:** 7
**Confidence:** 4

**Review:**

Protein surface geometry and chemistry play important roles in binding, but most existing surface-based methods only model local neighborhoods without considering long-range structural context. ProtSurf addresses this by proposing a hierarchical surface graph framework: a local encoder processes per-residue patch graphs to capture chemical and geometric features, while a global module incorporates backbone-level context across residues. The model is pretrained on large-scale protein structures using a surface reconstruction objective, and the resulting residue embeddings are used to augment GearBind for downstream affinity prediction on SKEMPI, HER2, and SAbDab.

---

**Pros:**
- The core idea of combining local surface geometry with global backbone context is well-motivated and reasonable.
- The two additional geometric features  (distance to $C_\alpha$ and a $C_\alpha$–$C_\beta$ angle) are simple but lead to a notable performance gain.

**Cons:**
- The overall framework involves many components (FSQ, Permuter, GRPE encoder/decoder), which seems a bit over-engineered. Some components like the Permuter lack ablation study.
- All downstream experiments are based on GearBind. It is not clear whether the learned representations are generally useful for other models.
- SAbDab training set only has around 100 samples, which makes the results less convincing. It would be helpful to evaluate on larger datasets like PDBbind.

**Questions:**
1. Section 4.1 reports Pearson $0.7003$ for ProtSurf (60M) but Table 1 shows $0.7033$ for the same model.
2. The Meaningfulness Statement mentions sequence and homolog-based methods, which does not match the actual content of this paper. It seems like it was written for a different project.

---

### Meta-Review · Area_Chair_eX4f · 2026-02-27

**Recommendation:** Accept (Poster)
**Confidence:** 4

**Metareview:**

Accept.

---

### Decision · Program_Chairs · 2026-03-02

**Decision:**

Accept (Poster)

**Comment:**

Please see the meta-review.